# Melatonin Protects Mitochondrial Function and Inhibits Oxidative Damage against the Decline of Human Oocytes Development Caused by Prolonged Cryopreservation

**DOI:** 10.3390/cells11244018

**Published:** 2022-12-12

**Authors:** Qi Zhu, Ding Ding, Han Yang, Weiwei Zou, Dandan Yang, Kaijuan Wang, Chao Zhang, Beili Chen, Dongmei Ji, Yan Hao, Rufeng Xue, Yuping Xu, Qiushuang Wang, Jing Wang, Bo Yan, Yunxia Cao, Huijuan Zou, Zhiguo Zhang

**Affiliations:** 1Department of Biomedical Engineering, Anhui Medical University, Hefei 230032, China; 2Department of Obstetrics and Gynecology, The First Affiliated Hospital of Anhui Medical University, Hefei 230022, China; 3NHC Key Laboratory of Study on Abnormal Gametes and Reproductive Tract, Anhui Medical University, Hefei 230032, China; 4Key Laboratory of Population Health Across Life Cycle, Anhui Medical University, Ministry of Education of the People’s Republic of China, No 81 Meishan Road, Hefei 230032, China; 5Anhui Province Key Laboratory of Reproductive Health and Genetics, Hefei 230032, China; 6Anhui Provincial Engineering Research Center of Biopreservation and Artificial Organs, Hefei 230032, China; 7The Second Clinical Medical School, Anhui Medical University, Hefei 230032, China

**Keywords:** melatonin, human oocytes, prolonged cryopreservation, development, mitochondrial function, oxidative damage

## Abstract

Melatonin (MT) can improve the effect of cryopreservation on oocytes by suppressing oxidative stress and maintaining the permeability of the oolemma. In this study, MT was firstly applied to human oocytes’ cryopreservation to explore the effect of prolonged cryopreservation on developmental competence and its role. Collected in vitro-matured human oocytes were cryopreserved in MT-containing or MT-free medium for 0 and 6 months; after warming, viable oocytes were assessed for developmental viability, intracellular protein expression, mitochondrial function, and oxidation-antioxidant system. Meanwhile, fresh oocytes were set as the control. The results showed that with the extension of cryopreservation time, the developmental competence of oocytes gradually declined, accompanied by the down-regulation of most mitochondrial function-related proteins, the reduction in ATP and GSH production, the increase in ROS accumulation, and the aggravation of the imbalance of ROS/GSH in oocytes. However, the participation of MT seemed to effectively mitigate these negative effects. Therefore, we speculate that melatonin may maintain normal ATP production and ROS/GSH balance in cryopreserved oocytes by protecting mitochondrial function and inhibiting oxidative damage, thereby effectively maintaining the developmental competence of human oocytes in prolonged cryopreservation.

## 1. Introduction

Since the birth of the world's first test-tube baby derived from the cryopreservation of human oocytes in 1986, this technology has received extensive attention in assisted reproductive technology [1]. Although human oocyte cryopreservation technology has been extensively developed, it has not been able to address major obstacles such as decreased developmental competence and uncontrollable survival rates, which means that it still not routinely applied in clinical practice [2,3]. The most probable reason is that the vitrification–warming process and cryopreservation cause unforeseen damage to the structure, metabolism, and developmental competence of oocytes [4,5,6]. Furthermore, long-term storage in liquid nitrogen reduced the fertilization and blastocyst formation rates of cryopreserved oocytes in rabbits and mice, while the survival rate and blastocyst formation competence of human embryos gradually decreased with the extension of cryopreservation time in humans [7,8,9]. However, it has also been shown that the duration of cryopreservation of human MII oocytes does not affect the gene expression profiles, either by slow freezing [10] or vitrification [11].

Melatonin (N-acetyl-5-methoxytryptophan, MT) is a type of indole compound, synthesized by the mammalian pineal gland, with high lipid solubility and low water solubility [12]. It is a powerful free radical scavenger and antioxidant [13,14] that has a wide range of physiological functions, including antioxidation [15], anti-aging [12], sleep promotion [16] and so on. MT and its metabolites can directly scavenge reactive oxygen species (ROS), activate antioxidant enzymes, increase the expression level and activity of Glutathione (GSH), regulate the expression of antioxidant genes, and inhibit prooxidative enzymes, thereby protecting cells from oxidative damage [17,18]. According to reports, the addition of 10^-9^ M MT to the culture medium may effectively improve the quality of mammalian gametes and embryos [19,20]. MT also plays an essential role in cryopreservation, and the application of MT may effectively protect mammalian gametes from oxidative stress during vitrification [13,21,22,23,24,25]. In 2021, Zhang et al. reported that MT could improve the effect of cryopreservation on human oocytes by suppressing oxidative stress and maintaining the permeability of the oolemma under short-term cryopreservation [15]. 

However, for the purpose of fertility preservation, gametes and embryos have to be cryopreserved for a long period, even for many years. Consequently, could prolonged storage at extremely low temperatures further impair the developmental competence of the samples? To date, few reports have systematically elucidated the effect of cryopreservation time on human oocytes. Therefore, based on our previous research [15], in the present study, in vitro-matured human oocytes (IVM-MII oocytes) with normal morphology were vitrified and stored in -196℃ liquid nitrogen for six months with the medium supplemented with 10^-9^ M MT, in order to systematically explore the effect of the cryopreservation time on the developmental competence of human oocytes, as well as the role of MT during cryopreservation.

## 2. Materials and Methods

### 2.1. Ethics Statement

This study was conducted at the First Affiliated Hospital of Anhui Medical University Reproductive Medicine Center and was approved by the Ethics Committee of Anhui Medical University (Ethics approval number: 20200114). Before conducting the study, the partners of all enrolled patients were interviewed and provided signed informed consent. 

### 2.2. Chemicals and Reagents

Unless otherwise specified, all reagents were purchased from Sigma Chemical Company (St. Louis, MO, USA). Additional details are provided in the Appendix A. 

### 2.3. Immature Oocyte Collection and In Vitro Maturation

The immature oocytes, including germinal vesicle (GV) and metaphase I (MI), were collected from 140 young infertile women (<35 years old). Then, in vitro maturation (IVM) was performed as described in our published literature [26], and high-quality IVM-MII oocytes were selected as the subjects for this study. Additional details are provided in the Appendix A.

### 2.4. Study Design

The collected high-quality IVM-MII oocytes were randomly divided into the following five groups: fresh group (F group, as control, n = 115), no-MT-treated cryopreservation for 0 months (nMC-0 group, n = 120), no-MT-treated cryopreservation for 6 months (nMC-6 group, n = 119), 10–9 M MT-treated cryopreservation for 0 months (MC-0 group, n = 121), 10-9 M MT-treated cryopreservation for 6 months (MC-6 group, n = 120). It is worth noting that 0 months means that oocytes were thawed immediately after vitrification, and 6 months means that oocytes were cryopreserved for 6 months before thawing. This study included three experiments. In experiment I, the vitrified–warmed oocytes in each group were subjected to ICSI insemination and cultured in vitro to observe the subsequent embryonic development. In experiment II, to further elucidate the mechanism by which MT inhibits the developmental competence degradation of human oocytes induced by prolonged cryopreservation, the micro-sample proteomics technique was used to detect the expression levels of proteins in individual oocyte. In experiment III, after cryopreservation, mitochondrial function was assessed by fluorescence staining and the levels of intracellular ATP, ROS, and GSH were detected in oocytes. The experimental flow chart is shown in Appendix A.

### 2.5. Oocyte Vitrification and Warming

Oocytes were vitrified and warmed according to our previous protocol [27]. After warming, morphological evaluation was performed under an optical microscope (IX-71, Olympus, Tokyo, Japan). The oocytes that were normal and round in shape, with moderate periocular space and clear separation of the cell membrane and cytoplasm, were considered viable. Additional details are provided in the Appendix A.

### 2.6. Developmental Competence

A total of 280 oocytes were subjected to ICSI insemination with donated sperm. The developmental competence of the oocytes, including fertilization, cleavage, blastocyst, high-quality blastocyst, etc., was assessed by the same embryologist under an optical microscope (IX-71, Olympus, Tokyo, Japan). Additional details are provided in the Appendix A.

### 2.7. Single-Oocyte Proteomic Assay

A total of 15 high-quality IVM-MII oocytes (5 per person) were collected from 3 young patients, and each patient’s oocytes were equally divided into 5 groups (F group (n = 1); nMC-0 group (n = 1); MC-0 group (n = 1); nMC-6 group (n = 1); MC-6 group (n = 1)). Oocytes under the five experimental conditions were prepared for a single-oocyte proteomic assay. Three parallel experiments were conducted for each group. Additional details are provided in the Appendix A.

### 2.8. Detection of ATP, ROS, and GSH Levels in Oocytes

A total of 94, 84, and 86 oocytes from the cryopreservation and F groups were further analyzed for ATP, ROS, and GSH levels, respectively. The detailed procedures are described in the Appendix A.

### 2.9. Statistical Analysis

Data were presented as mean ± standard deviation. Qualitative data were analyzed by chi-squared test or Fisher's exact test, while quantitative data were analyzed by one-way analysis of variance (ANOVA) and Bonferroni test. Statistical tests were performed using Prism version 8.0 (GraphPad, Insightful Science Inc., San Diego, USA). A *P*-value of 0.05 was considered statistically significant.

## 3. Results

### 3.1. Morphological Observation of Vitrified–Warmed Human Oocytes

A total of 480 vitrified–warmed oocytes were cultured for 2.5 hours after warming and observed under an optical microscope. In total, 440 (91.67%) oocytes were assessed as viable, of which 436 (90.83%) were selected for subsequent experiments. As shown in Figure 1, oocytes in the nMC-0 and nMC-6 groups showed various morphological changes compared with the F group, including vacuolization, rough cytoplasm or a large perivitelline space, and blurred membrane and cytoplasm separation. The nMC-6 group was found to present the roughest cytoplasm and the most severe vacuolization. In contrast, the oocytes in the MC-0 and MC-6 groups had uniform cytoplasm and no vacuoles. It is worth mentioning that the oocytes in the MC-0 group presented a similar morphology to fresh oocytes.

### 3.2. MT Protected the Developmental Competence of Human Oocytes in Prolonged Cryopreservation

To investigate the effect of prolonged cryopreservation on the developmental competence of human oocytes, a total of 280 IVM-MII oocytes were collected and randomly assigned to the following five groups: F group (*n* = 50), nMC-0 group (*n* = 58), MC-0 group (*n* = 60), nMC-6 group (*n* = 55), and MC-6 group (*n* = 57). Representative developmental images of fresh and vitrified–warmed oocytes after ICSI are shown in Appendix A. The developmental parameters (fertilization, cleavage, high-quality embryo, and blastocyst rates) of each group are shown in detail in Table 1. The results showed that the developmental competence of the no-MT-treated groups (2.5 h and 6 months) gradually decreased compared to the control group, especially for the fertilization, high-quality embryo, and blastocyst rates (Table 1). Similarly, for the MT-treated groups (2.5 h and 6 months), the developmental competence also showed a progressive decline, but there were no significant differences in each developmental parameter (Table 1). In addition, the developmental parameters of the MC-0 group were higher than those of the nMC-0 group, and there were significant differences in fertilization, high-quality embryo, and blastocyst rates (all *p* < 0.05); the developmental parameters of the MC-6 group were also higher than those of the nMC-6 group (Table 1). The above results suggest that the vitrification–warming process impairs the developmental competence of human oocytes, and the damage is aggravated with the extension of the cryopreservation time. Interestingly, the involvement of MT appears to inhibit the progressive degradation of developmental competence resulting from the vitrification–warming process and prolonged cryopreservation. 

### 3.3. Micro-Sample Global Proteomic Expression Characteristics of Human Oocytes in Prolonged Cryopreservation

To further analyze the mechanism of MT inhibiting the developmental competence degradation of human oocytes induced by prolonged cryopreservation, micro-proteomics was used to detect the protein expression level in individual oocyte. Fold change > 1.5 and *p* < 0.05 were determined as the screening criteria for significant differences among the five groups. A total of 6723 proteins were identified in this experiment, covering 1006 differential proteins (Figure 2A). Gene Ontology Functional (GO) enrichment analysis was conducted to identify the distribution of the expression of differentially expressed proteins between various biological functions (Figure 2B). The results showed that “cellular process” was the most significant enrichment term in the biological process category; “cell” was the most significant enrichment term in the cellular components category and “binding” was the most significant enrichment term in the molecular function category. 

Table 2 shows the number of total and mitochondrial-related differentially expressed proteins (DEPs) in the pairwise comparison. Figure 3 and Figure 4 show the biological functions in which these DEPs were involved. As shown Table 2, the GO-annotated DEPs statistics found that, compared with the F group, the nMC-0 group had 70 DEPs, with 64 downregulated and 6 upregulated (Appendix A), of which 14 mitochondrial function-related proteins were downregulated (Appendix A). The MC-0 group had 133 DEPs, including 129 downregulated and 4 upregulated (Appendix A), of which 20 mitochondrial function-related proteins were downregulated (Appendix A). Moreover, the nMC-6 group had 148 DEPs, including 144 downregulated and 4 upregulated (Appendix A), of which 21 mitochondrial function-related proteins were all downregulated (Appendix A). It is worth noting that the protein expression levels of the MC-6 group had almost no difference from those of the F group. There were only 8 upregulated and 9 downregulated proteins (Appendix A), of which only 2 mitochondrial function-related protein was upregulated (Appendix A). To analyze the contribution of MT, we compared the DEPs of the MT-treated group with the nMT-treated group at the same cryopreservation time. Compared with the nMC-0 group, the MC-0 group had 58 DEPs, of which 30 were downregulated and 28 were upregulated (Figure 3A), including 8 upregulated and 3 downregulated mitochondrial function-related proteins (Figure 4A). Compared with the nMC-6 group, the MC-6 group had 160 DEPs, with 4 downregulated and 156 upregulated (Figure 3B), of which mitochondrial function-related proteins were upregulated (Figure 4B). To analyze the effects of different cryopreservation times, we compared 0 months with 6 months of cryopreservation under the same intervention conditions. Compared with the nMC-0 group, the nMC-6 group had 53 DEPs, with 38 downregulated and 15 were upregulated (Figure 3C), of which one mitochondrial function-related protein was downregulated and one upregulated (Figure 4C), respectively. Compared with the MC-0 group, the MC-6 group had 212 DEPs, including 12 downregulated and 200 upregulated proteins (Figure 3D), of which 32 mitochondrial function-related proteins were upregulated (Figure 4D).

The common pathways responsible for the final oxidation by mitochondria are the tricarboxylic acid cycle and oxidative phosphorylation. Kyoto Encyclopedia of Genes and Genomes (KEGG) enrichment analysis revealed that the majority of DEPs involved in oxidative phosphorylation of the energy metabolism pathway were up-regulated in MC-6 group compared with nMC-6 group. Especially, most of these DEGs were encoded in the mitochondrial genome and participated in mitochondrial respiratory chain by encoding a component of vacuolar ATPase (V-ATPase). Those differentially expressed proteins included VATD, VATB2. Additionally, the genes encoding a subunit of mitochondrial ATP synthase including differentially expressed protein ATP5H. Mitochondrial ATP synthase catalyzes ATP synthesis, using an electrochemical gradient of protons across the inner membrane during oxidative phosphorylation. Cytochrome b-c1 complex subunit 1(QCR1), which is the key component of mitochondrial complex Ⅲ, were also found significantly up-regulated in the oocytes of MC-6 group (Table 3) (Figure 5). The results suggest that MT may be involved in the effect on mitochondria by regulating key proteins in the mitochondrial oxidative phosphorylation pathway and ATP synthesis.

### 3.4. MT Increased ATP Production of Human Oocytes in PROLONGED CRYOPReservation

As shown in Figure 6A,C, the level of intracellular ATP in the nMC-0 group (*n* = 20) or the nMC-6 group (*n* = 17) was significantly lower than that in the F group (*n* = 23) (*p* < 0.01; *p* < 0.001). Compared with the nMC-0 group, the ATP level in oocytes of the nMC-6 group was significantly decreased (*p* < 0.01). After MT treatment, the ATP level of the MC-0 group (*n* = 17) was significantly increased compared with that of the nMC-0 group (*p* < 0.05). Similarly, the ATP level of the MC-6 group (*n* = 17) was significantly increased compared with that of the nMC-0 group (*p* < 0.05). Similarly, the ATP level of the MC-6 group (*n* = 17) was significantly higher than that of the nMC-6 group (*p* < 0.01). Moreover, there was no significant difference in the ATP level between the MC-0 and MC-6 groups, which was comparable to that of F group. These results indicate that the ATP level of human oocytes dramatically decreased due to prolonged cryopreservation, and MT can effectively increase ATP production.

### 3.5. MT Inhibited the Aggravation of ROS Accumulation and Suppressed the Decline in GSH Production in Human Oocytes under Prolonged Cryopreservation

As shown in Figure 6B (upper panel) and Figure 6D regarding the ROS level in oocytes from each group, those in the nMC-0 (*n* = 17) and the nMC-6 groups (*n* = 17) were significantly higher than that in the F group (*n* = 19) (both *p* < 0.001), and the nMC-6 group showed a significantly higher fluorescence intensity of ROS (*p* < 0.05) compared with the nMC-0 group, which means that the freeze–thaw process and prolonged storage resulted in the dramatic accumulation of ROS; meanwhile, the result for the MC-0 group (*n* = 16) was significantly lower than that of the nMC-0 group (*p* < 0.001), and the result for the MC-6 group (*n* = 15) was significantly lower than that of the nMC-6 group (*p* < 0.001). As shown in Figure 6B (lower panel) and Figure 6E, regarding the GSH levels, those of the nMC-0 (*n* = 17) and nMC-6 (*n* = 17) groups were significantly lower than that of the F group (*n* = 20) (both *p* < 0.001), whereas the MC-0 (*n* = 16) and MC-6 groups (*n* = 16) were comparable to the F group. In addition, the MC-0 group’s value was higher than that of the nMC-0 group (*p* < 0.05), and the MC-6 group’s value was higher than that of the nMC-6 group (*p* < 0.05). These results suggest that the involvement of MT can inhibit the aggravation of ROS and suppress the decline in GSH production caused by the freeze–thaw process and prolonged storage.

### 3.6. ROS/GSH Ratio in Prolonged Cryopreserved Human Oocytes

Previous studies have demonstrated that the balance of the ROS/GSH ratio is critical for oocyte maturation and normal zygote development [28]. Therefore, we further analyzed the ROS/GSH ratio of vitrified–warmed human oocytes. As shown in Figure 6F, the ratio of ROS/GSH in oocytes of the nMC-0 and nMC-6 groups was significantly higher than that of the F group (both *p* < 0.001), and the ratio of the nMC-0 group was significantly lower than that of the nMC-6 group (*p* < 0.01). In addition, the ratio of the nMC-0 group was significantly higher than that of the MC-0 group (*p* < 0.01), and that of the nMC-6 group was significantly higher than that of MC-6 group (*p* < 0.001). In contrast, there was no significant difference in the ROS/GSH ratio between the MC-0 and MC-6 groups, and the ratio of each group was comparable to that of the F group. These results further suggest that the involvement of MT can reverse the changes in the ROS/GSH ratio induced by the freeze–thaw process and prolonged storage.

## 4. Discussion

Thus far, there are few reports on the effect of cryopreservation time on human oocytes. However, such research work has important clinical value for the preservation of human fertility. Therefore, combined with our previous findings on human oocyte cryopreservation, the present study was initiated to address two questions: (1) does prolonged cryopreservation aggravate the damage to the developmental competence of cryopreserved human oocytes? (2) Can the involvement of 10^-9^ M MT counteract the decline in developmental competence? 

Firstly, we investigated the effect of prolonged cryopreservation on the morphology and developmental competence of human oocytes. The vitrified–warmed oocytes in the no-MT treated groups showed various morphological changes; in particular, those in the nMC-6 group suffered the most serious damage. In contrast, the oocytes treated with MT had uniform cytoplasm and no vacuoles, and the morphological characteristics of the oocytes in the MC-0 group were almost indistinguishable from those of fresh oocytes. The morphological features point out that the vitrification-warming process has a negative effect on oocyte morphology, which is aggravated with the extension of the cryopreservation time, while the involvement of MT during cryopreservation significantly alleviated the negative effect. In subsequent developmental studies, the developmental competence of the oocytes in the no-MT-treated groups significantly decreased with the extension of the cryopreservation time. Moreover, the developmental competence of human oocytes in the MT-treated groups was significantly enhanced. The developmental results also suggest that the vitrification–warming process and a prolonged cryopreservation time will cause severe damage to oocytes, which can be effectively inhibited by MT intervention.

To reveal the potential pathogenesis of human oocyte degradation during cryopreservation, and to clarify the mechanism of MT’s resistance to cryogenic damage, micro-proteomics technology was utilized to detect the expression levels of proteins in a single oocyte. GO annotation revealed that the cellular process and metabolic process were the primary affected biological processes during oocytes’ cryopreservation. It was found that only the protein expression profile of the MC-6 group was almost comparable to that of the F group; most metabolic processes-related DEPs in other experimental groups were down-regulated. This suggests that cryopreservation adversely affects the metabolic function of oocytes. Mitochondria are the central site for ATP production in the metabolic process and ensure the provision of energy for oocyte and embryo development [29,30]. Cryopreservation has been reported to impair mitochondrial function and consume ATP [31]. Therefore, we further analyzed the mitochondrial function-related proteins. It was found that a considerable number of these downregulated DEPs were mitochondrial function-related proteins, which indicates that the mitochondrial proteins are sensitive to cryopreservation. Under the same intervention conditions, most mitochondrial function-related DEPs in the long-term preservation groups were down-regulated, which suggests that a prolonged cryopreservation time will impair the expression of mitochondrial function-related proteins. Under the same cryopreservation time, mitochondrial function-related DEPs in the MT-treated groups were all upregulated, which indicates that MT can preserve the expression of mitochondrial proteins and resist the negative effects of the vitrification–warming process and prolonged cryopreservation on mitochondria. Similar KEGG enrichment analysis showed that most DEPs involved in oxidative phosphorylation were upregulated in the MT-treated groups. Moreover, with the extension of the cryopreservation time, the addition of MT upregulated the expression of some proteins in the oxidative phosphorylation pathway.

The above results strongly demonstrate that the participation of MT can effectively preserve the expression of mitochondrial function-related and oxidative phosphorylation pathway-related proteins during human oocyte cryopreservation but cannot serve as direct evidence for the protection of mitochondrial function. One of the major functions of mitochondria is to produce ATP. Therefore, ATP production in cryopreserved oocytes was further measured. The ATP levels in oocytes of the nMC-0 group or nMC-6 group were significantly lower than that of the F group, indicating that the cryopreservation process reduces ATP production. It has been reported that the weakened mitochondrial function and impaired ATP production were present in human, porcine, and bovine oocytes after cryopreservation [31,32]. These findings are consistent with our findings. The ATP level in the nMC-6 group was significantly lower than that in the nMC-0 group, indicating that long-term cryopreservation results in a decrease in mitochondrial ATP production. However, the ATP levels in the MT-treated groups were significantly higher than those in the no-MT-treated groups, suggesting that MT can effectively protect mitochondrial function, thereby maintaining normal ATP production.

A by-product of energy production in mitochondria is ROS. Normally, overproduced cellular ROS is removed by the endogenous antioxidant system to maintain ROS homeostasis in cells. As the foremost nonenzymatic antioxidant, GSH is crucial for the formation, maintenance, and protection of meiotic spindles from oxidative stress [33]. The intracellular ROS/GSH balance is critically important for physiological functions of cells. An imbalance in ROS/GSH indicates that the cells are in a state of oxidative stress and suffer from oxidative damage [34]. According to recent reports, the vitrification–warming process and extremely low-temperature conditions destroy the ROS/GSH balance, resulting in the failed removal and accumulation of excessive ROS. The accumulated ROS will attack the target mitochondria, causing damage to the mitochondrial inner membrane and ultimately impairing mitochondrial function [35]. The damaged mitochondrial function further leads to the increased ROS and a disrupted ROS/GSH balance, resulting in the peroxidation of the biofilm of structural proteins and lipids and the impairment of the permeability of biofilms, finally causing cell apoptosis [35,36,37]. ROS/GSH balance is critical for promoting oocyte maturation and zygotic development [38]. Given that MT is known to exert antioxidant effects by modulating the intracellular ROS/GSH balance [39,40], we further detected the levels of ROS and GSH and analyzed the differences in ROS/GSH ratios among groups. In this study, the ROS and GSH levels of the no-MT-treated groups were significantly higher or lower than those in the F group, which indicated that cryopreservation resulted in ROS accumulation and GSH reduction. The ROS level of oocytes in the nMC-6 group was significantly higher than that in the nMC-0 group, confirming that ROS accumulation increased in oocytes in prolonged cryopreservation. After MT treatment, the ROS and GSH levels in the MC-0 and MC-6 groups were significantly diminished or enhanced compared to those in the nMC-0 and nMC-6 groups, respectively, which implies that the involvement of MT effectively suppresses the ROS accumulation and antioxidant competence impairment caused by prolonged cryopreservation. The ROS/GSH ratio in the no-MT-treated groups was significantly higher than that in the F and MT-treated groups, and that in the nMC-0 group was significantly lower than that in the nMC-6 group. However, the ROS/GSH ratio of the MT-treated groups was comparable to that of the F group. These results reveal that MT can effectively maintain the balance of the intracellular antioxidant system, thus effectively protecting the developmental competence of cryopreserved oocytes.

## 5. Conclusions

In conclusion, this study reveals the phenomenon of the decreased developmental competence of human oocytes with prolonged cryopreservation. The developmental competence degradation can be attributed to the deterioration of cellular metabolic function, especially mitochondrial function. MT can effectively protect the developmental competence of human cryopreserved oocytes. The underlying mechanism may be that melatonin protects mitochondrial function and inhibits oxidative damage mainly by protecting the expression of oxidative phosphorylation pathway-related proteins and maintaining ROS/GSH homeostasis, thereby maintaining normal ATP production and ROS/GSH ratio in human cryopreserved oocytes.

## Figures and Tables

**Figure 1 cells-11-04018-f001:**
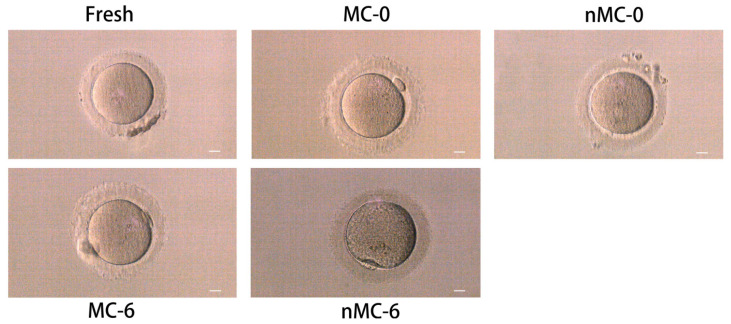
Effect of 10^-9^ M MT treatment on the effect of cryopreserved human oocytes from the nMC-0 group, MC-0 group, nMC-6 group and MC-6 group. The fresh group served as control. The morphological changes of oocytes were observed under an optical microscope: (1) vacuoles and granulation clusters in the cytoplasm of human oocytes in the nMC-0 group and nMC-6 group; (2) the coarsest cytoplasm and the most severe vacuolization in the nMC-6 group; (3) a large perivitelline space in the nMC-0 group; (4) normal morphology and uniform cytoplasm in the MC-0 group, MC-6 group and F group. Bar = 10 µm.

**Figure 2 cells-11-04018-f002:**
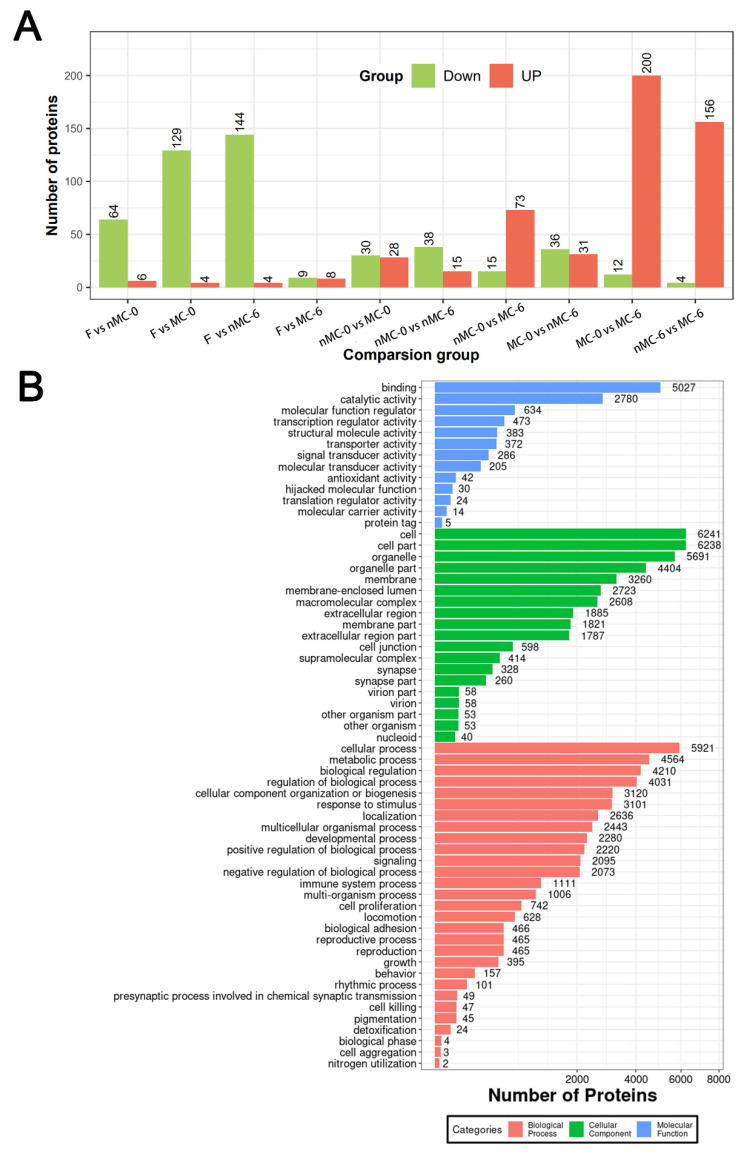
Effect of MT treatment and prolonged cryopreservation on the micro-sample global protein expression characteristics of cryopreserved human oocytes from the nMC-0 group, MC-0 group, nMC-6 group and MC-6 group. Fresh oocytes (F group) served as the control. (**A**) Differential Protein Statistics Graph. The histogram above is the statistical graph of differential proteins in each group. The green represents the number of down-regulated differential proteins, and the red represents the number of up-regulated differential proteins. (**B**) Gene Ontology (GO) Functional annotation. The GO classification map shows the distribution of the items involved in the three ontology, and the different colors are labeled as the individual items involved in the three ontology. The red represents biological processes, the green represents cellular component, and the blue represents molecular functions.

**Figure 3 cells-11-04018-f003:**
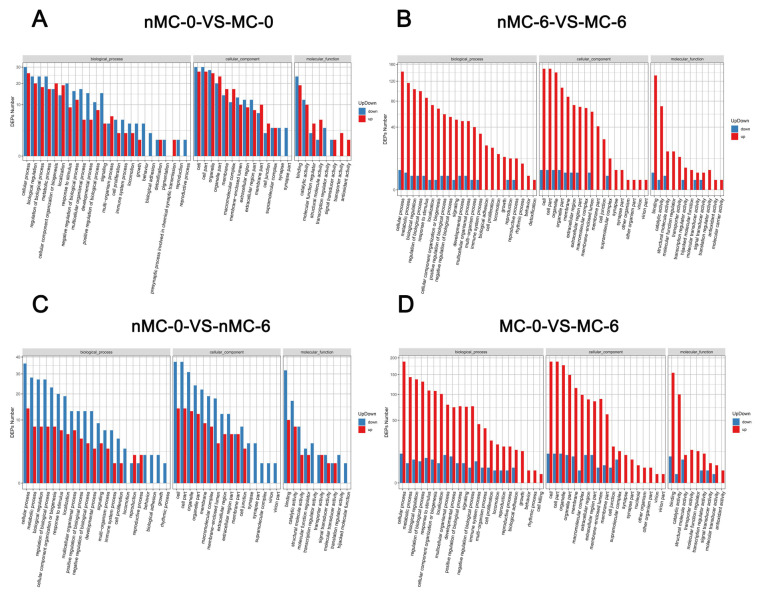
Differential protein GO function classification map of cryopreserved human oocytes from the nMC-0 group, MC-0 group, nMC-6 group and MC-6 group. (**A**) nMC-0 VS MC-0: Comparison of up-and down-regulated differential proteins between nMC-0 group and MC-0 group; (**B**) nMC-6 VS MC-6: Comparison of up-and down-regulated differential proteins between nMC-6 group and MC-6 group; (**C**) nMC-0 VS nMC-6: Comparison of up-and down-regulated differential proteins between nMC-0 group and nMC-6 group; (**D**) MC-0 VS MC-6: Comparison of up-and down-regulated differential proteins between MC-0 group and MC-6 group. The red represents up, the blue represents down.

**Figure 4 cells-11-04018-f004:**
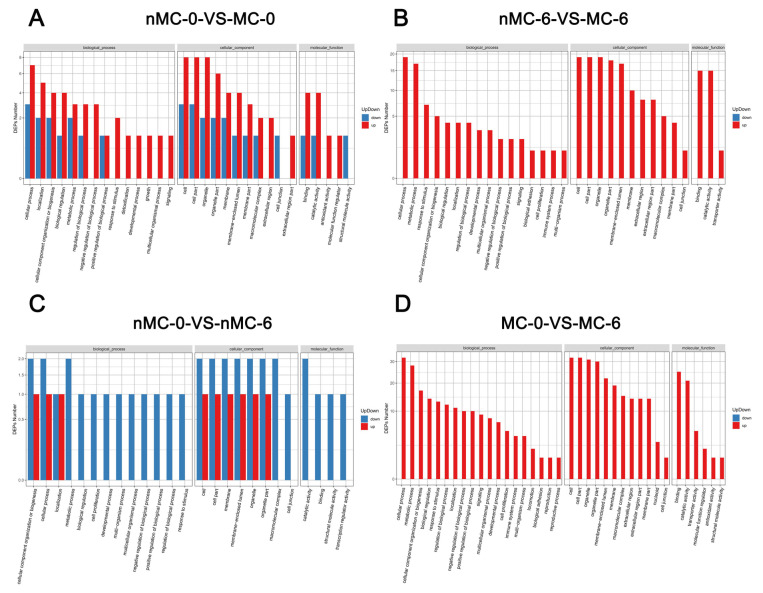
GO functional classification map of differential proteins associated with mitochondrial of cryopreserved human oocytes from the nMC-0 group, MC-0 group, nMC-6 group and MC-6 group. (**A**) nMC-0 VS MC-0: Comparison of up-and down-regulated differential proteins between nMC-0 group and MC-0 group; (**B**) nMC-6 VS MC-6: Comparison of up-and down-regulated differential proteins between nMC-6 group and MC-6 group; (**C**) nMC-0 VS nMC-6: Comparison of up-and down-regulated differential proteins between nMC-0 group and nMC-6 group; (**D**) MC-0 VS MC-6: Comparison of up-and down-regulated differential proteins between MC-0 group and MC-6 group. The red represents up, the blue represents down.

**Figure 5 cells-11-04018-f005:**
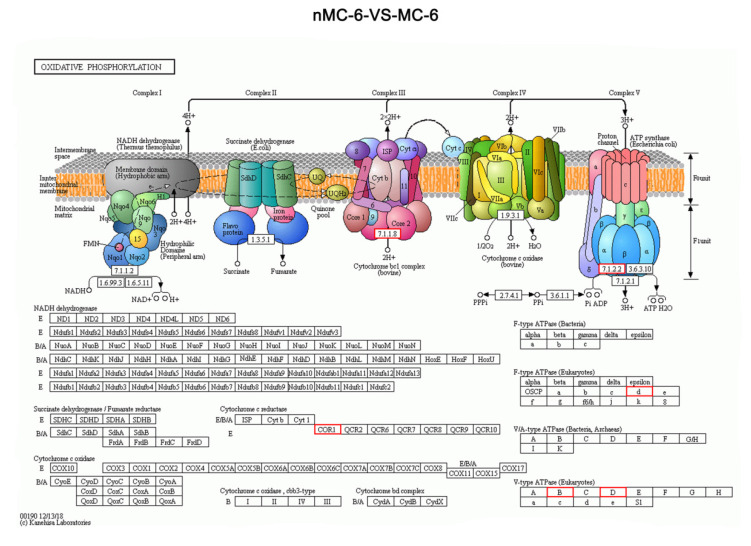
The KEGG pathway of oxidative phosphorylation pathway responds to cryopreserved human oocytes. The proteins highlighted in red are enriched and significantly upregulated in oocytes from the latter group. nMC-6 VS MC-6: Comparison of DEPs in oxidative phosphorylation pathway between nMC-6 group and MC-6 group.

**Figure 6 cells-11-04018-f006:**
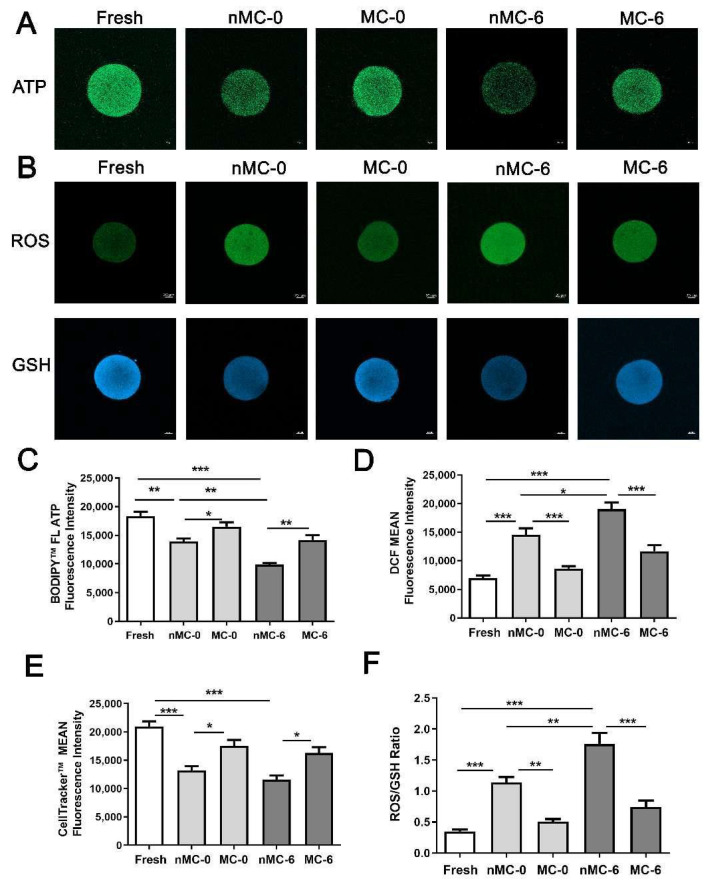
Effect of Effect of prolonged cryopreservation and 10^-9^ M MT treatment on mitochondrial function in cryopreserved human oocytes from the nMC-0 group, MC-0 group, nMC-6 group and MC-6 group. Fresh oocytes (F group) served as the control. (**A**)ATP levels in cryopreserved human oocytes. BODIPY^TM^ FL ATP staining was performed to measure the intracellular ATP levels, and the staining was detected using confocal microscopy. Bar = 20 µm. (**B**) ROS and GSH level in cryopreserved human oocytes. DCHFDA staining was performed to measure the intracellular ROS levels, CellTracker^TM^ staining was performed to measure the intracellular GSH levels, and the staining was detected using confocal microscopy. Bar = 20 µm. (**C**) BODIPY^TM^ FL ATP fluorescence intensity was quantified. Data were expressed as the mean ± SEM. * *p* < 0.05, ** *p* < 0.01 and *** *p* < 0.001. (**D**) DCHFDA fluorescence intensity was quantified. Data were expressed as the mean ± SEM. * *p* < 0.05 and *** *p* < 0.001. (**E**) CellTracker^TM^ fluorescence intensity was quantified. Data were expressed as the mean ± SEM. * *p* < 0.05 and *** *p* < 0.001. (**F**) Ratio of ROS/GSH in long-term cryopreserved human oocytes. The fresh group served as control. Data were expressed as the mean ± SEM. ** *p* < 0.01 and *** *p* < 0.001.

**Table 1 cells-11-04018-t001:** Effect of 10^-9^M MT treatment on the developmental competence of cryopreserved human oocytes.

	F	nMC-0	MC-0	nMC-6	MC-6
IVM-MII (n)	50	58	60	55	57
Fertilization rate (%)	86.00% (43/50) ^a,c^	62.07% (36/58)	83.33% (50/60) ^b^	58.18% (32/55)	64.91% (37/57)
Cleavage rate (%)	86.05% (37/43)	80.56% (29/36)	84.00% (42/50)	68.75% (22/32)	67.57% (25/37)
High-quality cleavage embryo rate (%)	64.86% (24/37) ^b,c^	34.48% (10/29)	61.90% (26/42) ^b^	18.18% (4/22)	48.00% (12/25)
Blastocyst rate (%)	51.35% (19/37) ^b,c^	17.24% (5/29)	40.48% (17/42) ^b^	9.09% (2/22)	28.00% (7/25)

Data were analyzed using the Chi-square or Fisher Exact test. Different symbols within columns and different letters within columns and within rows indicate significant differences.^a^
*p* < 0.01, compared with nMC-0 group; ^b^
*p* < 0.05, compared with nMC-0 group; ^c^
*p* < 0.01, compared with nMC-6.

**Table 2 cells-11-04018-t002:** Statistical table of the number of differential proteins related to mitochondrial function in each group.

	F vs. nMC-0	F vs. MC-0	F vs. nMC-6	F vs. MC-6	nMC-0 vs. MC-0	nMC-6 vs. MC-6	nMC-0 vs. nMC-6	MC-0 vs. MC-6
Number of differential proteins	70	133	148	17	58	160	53	212
Number of differential proteins associated with mitochondrial function	14	20	21	1	11	19	2	32
Up	0	0	0	2	8	19	1	32
Down	14	20	21	0	3	0	2	0

**Table 3 cells-11-04018-t003:** Statistical table of the number of differential proteins related to oxidative phosphorylation pathway between nMC-6 group and MC-6 group.

Primary_protein_ID	Mean Intensity (nMT-6)	Mean Intensity (MT-6)	Ratio	*p*-Value	Class	Description
p|Q9Y5K8|VATD_HUMAN	57,736,499.74	155,072,081.7	2.685858728	0.012634037	Up	V-type proton ATPase subunit D OS = Homo sapiens OX = 9606 GN = ATP6V1D PE = 1 SV = 1
sp|O75947|ATP5H_HUMAN	757,186,306.8	2,746,405,384	3.627119718	0.027221848	Up	ATP synthase subunit d, mitochondrial OS = Homo sapiens OX = 9606 GN = ATP5PD PE = 1 SV = 3
sp|P21281|VATB2_HUMAN	1,576,694,555	2,656,550,766	1.684886117	0.043175449	Up	V-type proton ATPase subunit B, brain isoform OS = Homo sapiens OX = 9606 GN = ATP6V1B2 PE = 1 SV=3
sp|P31930|QCR1_HUMAN	1,021,503,942	2,837,296,332	2.777567676	0.045073322	Up	Cytochrome b-c1 complex subunit 1, mitochondrial OS=Homo sapiens OX = 9606 GN = UQCRC1 PE = 1 SV = 3

## Data Availability

The data that support the findings of this study are available from the corresponding author upon reasonable request.

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
