# Peer review of "Melatonin Protects Mitochondrial Function and Inhibits Oxidative Damage against the Decline of Human Oocytes Development Caused by Prolonged Cryopreservation"

_cells, 2022, doi:10.3390/cells11244018_

Round 1

Reviewer 1 Report

In the manuscript entitle “Melatonin protects mitochondrial function and inhibits oxidative damage against the decline of human oocytes development caused by prolonged cryopreservation “ the authors described the effect of prolonged cryopreservation in aggravation of the developmental competence in human oocytes by deterioration of cellular metabolic function, especially mitochondrial function, and the possibility of preventing such effect by melatonin through protecting mitochondria-associated and oxidative phosphorylation related proteins, promoting ATP production and maintaining the balance of ROS/GSH. The manuscript expands a previous work published by the authors last year. For that reason, results are less novel since the ability of melatonin to protect the MC-0 group against nMC-0 has already been previously demonstrated.

Although experimental design is basically appropriate to try to address the hypothesis, some concerns should be considered.

·         The authors conclude that the protective effect of melatonin is due to a protective effect on mitochondrial function based on differences in ATP production and maintenance of the ROS/GSH ratio. However, this is a correlation of facts, which although may lead to that assumption but do not definitively prove it, so they should be more cautious when drawing conclusions.

·         The abstract, more than being a summary of the content of the manuscript, is a mere enumeration of results, so I recommend that it be rewritten.

·         I recommend a thorough revision of the English since there are numerous grammatical errors that make it very difficult to read and understand the manuscript.

·         Figure 1B is quite complex to understand. Perhaps the subdivision of the results in more than one figure could simplify their understanding. Also, shouldn't those data be represented with the corresponding error bars? Also, what is the difference between the data represented in Figure 1B and those represented in Table 1? It just looks like the same data represented in two different ways. Duplicities must be avoided.

·         The data represented in figures 3 and 4 are impossible to analyze since the texts in the figures cannot be read even when the magnification is maximized. Authors should pre-think such figures and find a representation that can be clearly viewed by readers. Similarly, figure 5 cannot be read clearly either. Although the representation of the mitochondrial transport loos very nice, I think it is totally unnecessary and that it only makes it difficult to interpret the results.

·         Regarding to the results represented in those figures, it looks like most of the protein changes at the mitochondrial level with melatonin occur at time 0. However, this is not what appears to be reflected in Table 2. Could the authors explain this fact better? Moreover, how is it explained that the changes observed in MC-0 are not present in MC-6? If the oocytes are cryopreserved, isn't the genetic machinery supposed to not work? How do you explain seeing differences in expressions between these two groups?

·         There are errors in the numbering of the figures that must be corrected.

Reviewer 2 Report

The article brings essential information about the use of Melatonin and the effect of preserving frozen oocytes for long periods, explaining the findings found

The MM is confused. I understand the need for supplementary material for the description of methodologies, but I believe that data from the experimental design should be detailed in the article.

The description of the donors was quite confusing. How many donors were used? Were their oocytes evenly divided between the experimental groups? Make these descriptions for each analysis performed

Supplementary material - requires the review of English, methodologies, and some reagents other than the concentration used

Results - Several points are MM and not results, for example, the number of donors, age...

Edit the file for Table 1 to be all on the same page

Describe the statistical analysis only in the item intended for this purpose

Round 2

Reviewer 1 Report

the authors have made some of the suggested changes and have responded to some of the questions raised. The writing of the abstract could still be improved, as well as the presentation of the results. Despite increasing the resolution of the figures, their visibility is still not good.

Author Response

Response to Reviewer 1 Comments

Point 1:

the authors have made some of the suggested changes and have responded to some of the questions raised. The writing of the abstract could still be improved, as well as the presentation of the results. Despite increasing the resolution of the figures, their visibility is still not good.

Response 1:

We sincerely thank for the reviewer’s positive comments. We have re-written the abstract according to your suggestion (Lines 30-38). The detailed description is as follows:

Abstract:

Melatonin (MT) can improve the effect of cryopreservation on oocytes by suppressing oxidative stress and maintaining the permeability of the oolemma. In this study, MT was firstly applied to human oocytes’ cryopreservation to explore the effect of prolonged cryopreservation on developmental competence and its role. Collected in-vitro-matured human oocytes were cryopreserved in MT-containing or MT-free medium for 0 and 6 months; after warming, viable oocytes were assessed for developmental viability, intracellular protein expression, mitochondrial function, and oxidation-antioxidant system. Meanwhile, fresh oocytes were set as the control. The results showed that with the extension of cryopreservation time, the developmental competence of oocytes gradually declined, accompanied by the down-regulation of most mitochondrial function-related proteins, the reduction of ATP and GSH production, the increase of ROS accumulation, and the aggravation of the imbalance of ROS/GSH in oocytes. However, the participation of MT seemed to effectively mitigate these negative effects. Therefore, we speculate that melatonin may maintain normal ATP production and ROS/GSH balance in cryopreserved oocytes by protecting mitochondrial function and inhibiting oxidative damage, thereby effectively maintaining the developmental competence of human oocytes in prolonged cryopreservation.

As per your suggestion, we also revised the results. we have deleted some contents as follows:  “and its potential mechanism during cryopreservation” (line 151); “; the F group’s results were significantly higher than those of the nMC-0 group (P<0.01; P<0.05; P<0.05) and nMC-6 group (all P<0.01)” (line 158) and “Figure 3 and Figure 4 show the statistical graph of the relationship between the upregulation and downregulation of total and mitochondrial-related differentially expressed proteins (DEPs) in GO functional classification.” (line 201).

We apologize for the resolution of the figures in the manuscript, and therefore we have improved the figures 3, 4 and 5 and to make the figures easier to read. The results of this part have also been revised accordingly (line 213-258). Another part can be seen in the figures S3 and S4 we added in the Supplementary Data of the revised version.

In the revised manuscript, we adjusted the presentation as shown in conclusion (line 431-435): The underlying mechanism may be that melatonin protects mitochondrial function and inhibits oxidative damage mainly by protecting the expression of oxidative phosphorylation pathway-related proteins and maintaining ROS/GSH homeostasis, thereby maintaining normal ATP production and ROS/GSH ratio in human cryopreserved oocytes.
